# Direct Integration of 3D Printing and Cryogel Scaffolds for Bone Tissue Engineering

**DOI:** 10.3390/bioengineering10080889

**Published:** 2023-07-27

**Authors:** Levi M. Olevsky, Amritha Anup, Mason Jacques, Nadia Keokominh, Eric P. Holmgren, Katherine R. Hixon

**Affiliations:** 1Thayer School of Engineering, Dartmouth College, Hanover, NH 03755, USA; levi.olevsky.th@dartmouth.edu (L.M.O.); amritha.anup.th@dartmouth.edu (A.A.); 2College of Engineering and Physical Sciences, University of New Hampshire, Durham, NH 03824, USA; mason.jacques@dartmouth.edu (M.J.); nadia.keokominh@unh.edu (N.K.); 3Geisel School of Medicine, Dartmouth College, Hanover, NH 03755, USA; eric.p.holmgren@hitchcock.org

**Keywords:** tissue engineering, cryogel, 3D printing, scaffold, gyroid, bone graft substitute, bone healing

## Abstract

Cryogels, known for their biocompatibility and porous structure, lack mechanical strength, while 3D-printed scaffolds have excellent mechanical properties but limited porosity resolution. By combining a 3D-printed plastic gyroid lattice scaffold with a chitosan–gelatin cryogel scaffold, a scaffold can be created that balances the advantages of both fabrication methods. This study compared the pore diameter, swelling potential, mechanical characteristics, and cellular infiltration capability of combined scaffolds and control cryogels. The incorporation of the 3D-printed lattice demonstrated patient-specific geometry capabilities and significantly improved mechanical strength compared to the control cryogel. The combined scaffolds exhibited similar porosity and relative swelling ratio to the control cryogels. However, they had reduced elasticity, reduced absolute swelling capacity, and are potentially cytotoxic, which may affect their performance. This paper presents a novel approach to combine two scaffold types to retain the advantages of each scaffold type while mitigating their shortcomings.

## 1. Introduction

Bone disorders caused by infection, trauma, or tumor resection are highly prevalent, highlighting the need for improved treatments to induce bone regeneration; current treatment for a defect less than 6 cm is bone grafting [1,2]. Harvesting bone from a donor site can lead to associated complications such as pain, infection, and nerve damage [3,4]. Additionally, the limited supply of donor tissue can be a significant challenge, particularly in cases requiring multiple grafts or repeat surgeries [3]. Furthermore, bone grafts may fail to fully integrate with the surrounding tissue, leading to poor mechanical stability and the need for additional surgeries [5]. These limitations have motivated the development of alternative treatments for bone regeneration [6]. These novel approaches aim to address the limitations of bone grafting by providing a biocompatible and mechanically stable environment for bone tissue regeneration while promoting the differentiation and proliferation of bone cells [6].

Tissue engineering is an interdisciplinary field that involves the integration of biomaterial scaffolds, cells, and bioactive factors to promote targeted growth and regeneration of new tissue [7]. Therefore, the implementation of a tissue-engineered scaffold framework that supports cell proliferation, migration, and attachment could present a promising substitute for bone grafting. While the clinical and economic advantages of tissue engineering are recognized, there are still areas that require attention to enhance translation from bench to bedside [6,7]. In particular, the optimization of patient-specific biomaterials to mimic physical properties of bone represents a significant challenge [6]. Specifically, bone tissue possesses unique mechanical properties, including stiffness, strength, and toughness, which enable it to support and protect the body [8].

Previous studies have explored various scaffold fabrication techniques to target bone formation [5,6,7,9,10,11,12,13,14]. Cryogel scaffolds are produced by the freezing and subsequent thawing of a polymer solution, resulting in a sponge-like, macroporous structure that is ideal for cellular infiltration and angiogenesis [14]. Further, natural materials including chitosan and gelatin can be incorporated to increase biocompatibility, biodegradability, and non-toxicity [10]. Chitosan–gelatin cryogels are a popular choice for bone tissue engineering due to their biocompatibility and ideal physical properties such as pore size, swelling potential, and compressive moduli [10,14,15,16,17,18,19,20,21,22,23]. Cryogels possess a high modulus of resilience, allowing them to be highly compressed without permanent deformation; however, they have a low modulus of elasticity, making these scaffolds mechanically weak and unable to bear high loads [14]. This creates complications in using these scaffolds for large defects, where no cryogels have been used clinically. Comparatively, 3D printing is another method for producing tissue-engineered scaffolds as it allows for the detailed printing of patient-specific geometries [24]. This method provides a wide range of options for material selection, enabling the customization of scaffolds to achieve desired properties; in this case, a desired property is increased mechanical strength [24]. Despite these advantages, the limited printing resolution of commonly available 3D printers remains a drawback in achieving the microstructure necessary for adequate tissue regeneration, where ideal pore sizes range from pore diameters of 100 to 200 µm [24,25]. The resolution of commonly used 3D printers ranges from 70 to 250 µm [20]. Therefore, there are many advantages to directly combining these two scaffold fabrication methods to create an integrated, mechanically strong scaffold that promotes cellular adhesion. Although cryogel scaffolds have been produced via 3D printing, as well as fabricated inside disposable molds, the direct integration of cryogel fabrication with 3D printing has yet to be explored [13].

To appropriately combine these two fabrication methods, the 3D-printed framework must interface with the cryogel, while increasing the mechanical strength of the combination scaffold and supporting cellular adhesion and proliferation. The gyroid shape has recently gained attention as a lattice framework for tissue engineering scaffolds [9,26,27,28]. Specifically, the unique shape has a highly interconnected porous structure, providing a large surface area for cell attachment and proliferation, efficient transport of nutrients and waste, and controlled mechanical properties [9,27]. The gyroid lattice is composed of repeating triply periodic minimal surfaces, which can be generated through mathematical algorithms and fabricated using advanced manufacturing techniques, such as 3D printing [29]. This structure can be tuned to match the mechanical properties of the surrounding tissue and promote cell differentiation by varying the strut thickness, pore size, and overall dimensions. The gyroid shape can also provide a favorable microenvironment for cell attachment, proliferation, and differentiation, as well as angiogenesis, due to its high surface area-to-volume ratio [30]. Overall, the gyroid shape has demonstrated great potential for tissue engineering applications, specifically in bone tissue engineering, where it can promote bone regeneration and improve implant integration.

In this study, we directly combined the two scaffold fabrication methods (cryogels and 3D printing) to create a combined structure with high porosity to ensure healthy cellular growth, while increasing mechanical durability to applied loads by the surrounding in vivo tissues. We hypothesize that combining cryogel scaffolds with 3D-printed gyroids will provide a mechanically stable, macroporous structure to support the creation of patient-specific scaffolds for complex bone defects. The null hypothesis assumes that there is no significant difference between the properties of the combined scaffolds compared to the control cryogels. Conversely, the alternative hypothesis suggests that the combined scaffolds, when compared to the control cryogels, will have smaller average pore sizes, lower absolute and relative average swell potential, and higher average compressive moduli.

## 2. Materials and Methods

### 2.1. Gyroid Lattice Design

MSLattice was used to generate a solid cuboid gyroid sample (Figure 1) [31]. The program accepts six parameters: width, length, height of the sample (in unit cells), unit cell size (dimensionless), relative gyroid density of the sample (0–100%), and mesh density points. The generated gyroid samples used 10%, 20%, and 30% relative densities and 2.0, 2.5, and 3.0 unit cell sizes. Mesh density points were kept constant at 50, and sample length, width, and height were set to an arbitrary value to generate a larger gyroid than was needed in every dimension. The cuboid gyroid STL files generated by MSLattice were then imported into Blender (Blender Institute; Amsterdam, Netherlands) using a unit conversion of 1 unit = 1 mm, where a Boolean intersection was performed between them and a cylinder (10 mm height by 9 mm diameter). Unconnected mesh artifacts from this procedure were removed within Blender. The cylinders were then scaled down to 8.9 mm diameter, to fit within the syringes. In all, nine lattices were generated with varying relative density (10%, 20%, 30%) and pore size (2.0, 2.5, 3.0 mm; Figure 1). Relative density refers to the solid volume fraction of the lattice which is defined as the ratio of the solid volume of the lattice to the volume of the space that the lattice occupies [31]. Relative density will be referred to as solidity henceforth as this is a more intuitive way to describe the 3D-printed lattices; higher solidity lattices use more material to manufacture and possess thicker branches. STL files of the lattices can be downloaded from the Appendix A.

### 2.2. 3D Printing Gyroid Lattice

The 3D-printed lattices were printed on a Form 3B SLA 3D printer (Formlabs, Massachusetts) using Grey V4 Resin (Formlabs, Somerville, MA, USA) and Preform 3D print preparation software (Formlabs, Somerville, MA, USA). The lattices were printed using a 0.160 mm layer height and Preform’s Default (v1.1) print settings. The lattices were printed as cylinders with heights of 10 mm and diameters of 9 mm (Figure 2).

### 2.3. Preparation of Chitosan–Gelatin (CG) Solution

CG cryogels were made with reference to previously described methods [10]. An aliquot of 10 mL of 1% acetic acid (Fisher Scientific, Fair Lawn, NJ, USA) in deionized water (DI) was prepared. This solution was then split into 8 and 2 mL scintillation vials. Low-viscosity chitosan (80 mg, Mw = 1526.464 g/mol; MP Biomedicals, Solon, OH, USA) was added to the 8 mL aliquot and vortexed for 30 sec before placing it on a mechanical spinner for 1 h. Gelatin from cold water fish skin (320 mg, Mw = 60 kDa; Sigma-Aldrich, St. Louis, MO, USA) was then added to the 8 mL aliquot and placed on a mechanical spinner for 1 h, ensuring the gelatin was completely dissolved. The remaining 1% acetic acid 2 mL vial was combined with glutaraldehyde (Sigma-Aldrich, St. Louis, MO, USA) to create a 1% glutaraldehyde solution. The 8 and 2 mL scintillation vials were then placed in a 4° fridge for 1 h.

### 2.4. Combined Scaffold Integration

To start, 3 mL syringes (Fisher Scientific, Fair Lawn, NJ, USA) containing two gyroids each were pre-frozen at −23 °C for 6 h. The 8 mL and 2 mL solutions described previously were then mixed by decanting between the vials and immediately poured into the syringes. Once the syringes were filled, the plunger was inserted and the syringe was flipped upside down. The plunger was depressed until the cryogel solution completely infiltrated the entire gyroid, allowing for some cryogel solution to be expelled from the syringe if necessary. Immediately following this, the syringes were placed in a −23 °C freezer for 18 h to crosslink at subzero temperatures. Figure 3 shows the final combined scaffolds thawing in the syringes. Plain cryogels served as the control and will be referred to as control cryogels henceforth (Figure 2).

### 2.5. Pore Analysis

Scanning electron microscopy (SEM; VEGA3; TESCAN, Brno, Czech Republic) was used to observe the pore structure on the cryogels. Combined scaffolds were frozen at −80 °C for 1 h prior to being lyophilized (FreeZone Freeze Dryer, Labconco, Kansas City, MO, USA) overnight. The samples were then mounted on an aluminum stub and sputter coated (HUMMER 6.2; Anatech, Sparks, NV, USA) for 240 s in gold at 15 mA under the pulse setting to avoid overheating. SEM was then used to obtain images at 100, 200, 500, and 1000× or all combined scaffolds.

ImageJ was used to analyze the pore diameter in the combined scaffolds. First, the line function was selected and was used to determine the scale via the scale bar in the SEM image. The unit of length was adjusted to microns for accurate measurements. Next, the selection tool was used to measure the length of a representative pore, specifically focusing on the long diameter. The measurement was recorded and the process was repeated 60 times, taking 15 measurements from each quadrant of the image. The data were saved in an Excel file and the length values were used for statistical analysis.

### 2.6. Swelling Kinetics

To evaluate the shape retention and rehydration potential of combined scaffolds, a swelling test was performed. Three samples of each scaffold variation were lyophilized for 24 h prior to recording the dry weight. Each sample was placed in a weigh boat containing 5 mL DI water, removed, and weighed at nine time points: 2, 4, 10, 20, 40 min, 1, 2, 4, and 24 h. The average swelling ratio, considering the original dry weight of each sample, was recorded using the equation [10]:(1)$$Swelling\;ratio=(W_{h}-W_{d})/W_{d},$$
where *W_h_* is the hydrated weight and *W_d_* is the dry weight.

The weight of the plastic gyroid was subtracted from the final weights to isolate the swelling that is derived from only the cryogel portion of the combined scaffold.

### 2.7. Mechanical Testing

Ultimate compression testing was conducted to 75% for all combined scaffold cryogels following hydration for 5 min in phosphate buffered saline (PBS; Fisher Scientific; Fair Lawn, NJ, USA). Compression was completed using an Intron 68SC-2 system (Instron, Norwood, MA, USA) with a 500 N load cell and set parameters of a test rate of 10 mm/min, preload of 0.05 N, and preload speed of 1 mm/min. All data were analyzed through the Bluehill universal software (Instron; Norwood, MA, USA), and the compressive modulus (MPa) was taken from the software output for the combined scaffolds and control cryogels.

### 2.8. Cellular Infiltration

Based on these results, a total of 15 samples (*n* = 3 per time point) of each of the 20% solidity combined scaffolds were sterilized in 70% ethanol (Fisher Scientific; Fair Lawn, NJ, USA) for 30 min, followed by three 10 min washes with sterile PBS. The combined scaffolds were then placed in a sterile 24-well plate (Falcon, Marlboro, NY, USA) with 100 μL of Dulbecco’s Modified Eagle’s Medium (DMEM; ThermoFischer Scientific, Waltham, MA, USA), 10% fetal bovine serum (Omega Scientific Inc.; Tarzana, CA, USA), and 1% penicillin–streptomycin solution (Life Technologies Corporation, Carlsbad, CA, USA) containing 50,000 human bone osteosarcoma-derived cells (MG-63, passage 97; ATCC, Manassas, VA, USA). The cells were seeded on each scaffold through a dropwise method and left to incubate for two hours at 37 °C and 5% CO_2_ to allow for cell attachment. After this time period, an additional 200 μL of complete media was added so that all samples were fully submerged. Media was changed every two to three days from around the scaffold. The combined scaffolds and media were cultured to day 5, 7, 14, 21, or 28, at which time the combined scaffolds were placed in formalin (Fisher Scientific; Fair Lawn, NJ, USA) for 24 h and then stored in PBS.

To prepare the cryoprotectant medium, a 30% sucrose (*w*/*v*; Sigma-Aldrich, St. Louis, MO, USA) solution in DI was created and thoroughly mixed on a shaker plate. The cryogels, previously stored in PBS, were then submerged in individual 5 mL Eppendorf tubes filled with the cryoprotectant solution. These tubes were placed in a 4 °C refrigerator for 24 h. After the cryoprotection process, a gelatin–sucrose embedding solution was prepared [32]. A 5% gelatin from porcine skin (*w*/*w*; Sigma-Aldrich, St. Louis, MO, USA)–5% sucrose (*w*/*w*; Sigma-Aldrich, St. Louis, MO, USA) solution in DI water was prepared and dissolved using a water bath. Embedding molds were retrieved and appropriately labeled. Once the embedding solution was fully prepared, 1 mL of the solution was added to each mold and the cryoprotected samples were transferred to their respective molds. Additional embedding solution was added to ensure complete coverage of the scaffolds. The molds were then placed in a 45 °C oven and incubated for 2 h. The samples were then transferred to a −80 °C freezer and allowed to freeze overnight. Cryosectioning (Cryostat Microm HM 525; Thermo Scientific, ThermoFischer Scientific, Waltham, MA, USA) was performed at a thickness of 20 μm, and the sections were stained with 4′,6-diamidino-2-phenylindole (DAPI; BD Biosciences, San Diego, CA, USA) to assess cellular characteristics, as previously described in published work [33]. Images were taken by an optical light microscope (Laxco, Mill Creek, WA, USA) at 100×.

### 2.9. Statistics

GraphPad Prism was utilized to conduct all statistical analyses, employing a significance level of 0.05. A two-way ANOVA, followed by a Tukey post hoc analysis, was performed to assess the significance among different groups. The presence of outliers in the swelling ratio was determined using the ROUT method, with a Q value of 1%.

## 3. Results

### 3.1. Pore Analysis

SEM was used to capture images of the combined scaffolds, as shown in Figure 4. The pore areas for each scaffold were calculated using ImageJ. The resulting average and standard deviations (std. dev.) for the nine types of combined scaffolds are presented in Table 1. The distribution of pores can be seen in Figure 5. There were no significant differences between any of the scaffolds or the control.

### 3.2. Swelling Kinetics

Following dehydration and subsequent immersion in water, all combined scaffolds achieved maximum swelling capacity in 2 min (Figure 6). To accurately assess the swelling ratio of the cryogel within the combined scaffold, the weight of the plastic lattice was subtracted from the overall weight of the combined scaffold; this ensured that only the weight of the cryogel was considered in each final measurement. Post-processing analysis revealed that the 10%, 20%, and 30% solidity combined scaffolds swelled to approximately 1800–1900%, 1500–1800%, and 1000–1200% of their respective dry weights. There was no significant difference in swelling ratio for the control cryogel after the 4 min mark and no significant difference in swelling ratio after the 2 min mark for each combined scaffold (Q = 1%). However, the swelling ratios of the 30% solidity combined scaffolds were significantly different from the control group at all time points. Additionally, the swelling ratios of the 10% 2.0 mm combined scaffolds exhibited significant differences when compared to the 10% 3.0 mm combined scaffolds at all time points, except for the 20 min mark (*p* < 0.05).

### 3.3. Mechanical Testing

The ultimate compression of the scaffolds was assessed at 75% strain. The stress vs. strain graphs for the 10% solidity combined scaffolds exhibited a steady increase, whereas the stress vs. strain graphs for the 20% solidity and 30% solidity combined scaffolds exhibited multiple peaks (Figure 7). The 30% solidity combined scaffolds demonstrated the highest compressive moduli, followed by the 20% solidity and 10% solidity scaffolds, respectively. The average compressive modulus of the control was 0.032 MPa with a standard deviation of 0.0029 MPa. The solidity of each combined scaffold was significantly different from that of the control group (*p* < 0.05). Furthermore, there were significant differences in the compressive modulus between the scaffolds with 10% solidity and the scaffolds with 30% solidity (*p* < 0.05). Similarly, the compressive modulus of the scaffolds with 20% solidity differed significantly from that of the scaffolds with 30% solidity (*p* < 0.05). The compressive moduli for each scaffold are presented in Table 2 and visualized in Figure 8.

### 3.4. Cellular Infiltration

All combined scaffolds were seeded with MG-63 osteosarcoma cells and incubated to support infiltration over 28 days. Cryosectioning was used to obtain cross-sectional samples from five different regions within each scaffold to assess the extent of cellular infiltration (Figure 9).

## 4. Discussion

The objective of this study was to assess the advantages of incorporating a 3D-printed lattice into a cryogel scaffold. Nine distinct 3D-printed lattices were fabricated by varying the pore size (2.0, 2.5, 3.0 mm) and solidity (10, 20, 30%) of a gyroid shape. Subsequently, the nine lattice structures were permeated with cryogel, producing nine unique sample types to assess the novel combined scaffold. Note that the final diameter of the combined scaffolds was 8.5 mm instead of 9.0 mm due to slight breaking of the plastic lattices as they were inserted into the syringes; the decrease in diameter did not negatively affect any of the experiments. The combined scaffolds were evaluated and compared with control cryogel based on porosity, swelling capacity, mechanical integrity, and cell compatibility.

Visual examination of the SEM images in Figure 4 demonstrates that the incorporation of a 3D-printed plastic lattice into the cryogel scaffold impacts the pore architecture. Specifically, there appears to be some interaction between the cryogel and lattice material, as evidenced by attachment points between them. Interestingly, SEM analysis indicated that the pores farther from the lattice appeared larger than those in the control cryogel, while those closer to the attachment points were smaller. This pattern may be explained by the plastic’s hinderance of ice crystal formation within the cryogel. The ice crystals near the plastic lattice would hypothetically be smaller than the ice crystals further from the lattice. As a result, the smaller ice crystals melt to leave behind smaller pores near the lattice and larger ice crystals melt to produce bigger pores further from the lattice. The pores also exhibit spherical or ovoid shape similar to human trabecular bone structure demonstrated in previous studies [21,22]. Despite this, the addition of the plastic lattice did not significantly alter the pore size of the combined scaffolds, except in the 10% 2.5 mm sample. It is crucial to maintain the porosity of the cryogel to ensure its successful incorporation into bone defect sites, as pore size and interconnectivity are essential for cell attachment and the transport of nutrients and growth factors [25]. Previous research suggests that a pore diameter of 100–200 μm is optimal for bone regeneration [25]. Although the mean pore sizes in this study were below the optimal range, they remained similar to the control cryogel. We hypothesize that the pore size for all samples shrunk due to lyophilization, as cellular infiltration was achieved (discussed below). It is also possible that the hydrated form of the scaffold could have larger pores. Future research should investigate the pore size in its hydrated state using microCT [34].

The ability to rapidly swell is an advantageous property for scaffolds in bone tissue engineering applications [35]. High swelling capacity allows cryogels to quickly fill bone defect sites without the need for pre-wetting, facilitating nutrient absorption and promoting even cell distribution for enhanced regrowth at wound sites [36]. In this study, all cryogels achieved maximum swelling capacity after four minutes. However, incorporation of a 3D-printed lattice with 30% solidity reduced the swelling capacity compared to control cryogel. While the 10% and 20% solidity combined scaffolds exhibited no significant difference in swelling capacity compared to control cryogel, their absolute swelling capacity was also lower (Figure 6). Therefore, the presence of a 3D-printed lattice that does not swell may hinder nutrient absorption and decrease the number of infiltrating cells in the overall scaffold in vivo.

Cryogels possess a highly porous, sponge-like structure with remarkable elastic properties, allowing them to be compressed to 90% and rebound to their original shape without experiencing crack propagation [37]. Compression testing was performed to evaluate the mechanical durability and strength of the combined scaffolds. The stress–strain curves for the 10% solidity combined scaffolds showed a gradual increase in slope, indicating a lack of abrupt scaffold fracture with increasing strain (Figure 7). Conversely, the stress–strain curves for the 20% and 30% solidity combined scaffolds exhibited multiple peaks, corresponding to different lattice layers breaking under high loads (Figure 7). Our findings show that increasing lattice solidity resulted in an increased compressive modulus for the combined scaffolds, likely due to the greater proportion of plastic in the combined scaffold resulting in an improved ability to resist compression (Figure 8). Notably, the 2.5 mm lattices exhibited the highest mechanical durability of all lattice solidities, potentially because of a balance between the stiffness from the plastic and elasticity from the cryogel. It is interesting to note that as solidity increased, variability of compressive modulus increased (Figure 8). This may be due to a greater amount of cryogel in the lower solidity scaffolds compressing more elastically, whereas the higher solidity scaffolds had more plastic which had more random fracture patterns. These results suggest that the combined scaffolds possess improved mechanical properties compared to control cryogels, supporting their use in bone defects that may experience greater relative loading.

For tissue to grow into the cryogel, it is crucial for cells to infiltrate and proliferate within it. Only the three 20% solidity combined scaffolds were used for the cell infiltration study in order to reduce the number of scaffolds that would have to be created to assess all nine scaffold types. The 20% solidity combined scaffolds were chosen because they possessed favorable properties including similar pore size and relative swell capacity to the control, as well as a medium absolute swell capacity and compressive modulus compared to the 10% and 30% scaffolds. Further, using three different gyroid pore sizes (2.0, 2.5, 3.0 mm) allowed for an investigation in the effect on cell proliferation through the super-macropores of the 3D-printed scaffold portion. In the first five days, MG-63 cells exhibited adherence and proliferation across all three combined scaffolds, with cells distributed throughout the top layer and penetrating into the middle of the scaffolds. However, a reduction in cell count was observed between days 7 and 14, and by days 21 and 28, cells were solely detected on the surface or were absent entirely. Per ISO 10993-5 Section 8.5.1, a reduction in cell viability by more than 30% is considered a cytotoxic effect [23]. Because the combined scaffolds showed close to 100% reduction in cell viability as determined by an absence of visible cell nuclei, the combined scaffolds are determined to be cytotoxic. This cytotoxic effect could be attributed to either the composition of the plastic resin or the leaching of substances from the resin. Formlabs’s Grey Resin contains urethane dimethacrylate, which has been shown to be cytotoxic to MG-63 cells [38,39]. Among the different scaffolds, the 20% 3.0 mm scaffold exhibited the highest number of cells for the longest period of time (up to day 14). This can be attributed to the larger volume of cryogel it contained, as well as the presence of fewer obstacles that the cells needed to navigate through. It was difficult to determine if cells had adhered to the plastic surface, as the plastic exhibited significant fluorescence. Additionally, imaging the samples posed challenges due to the plastic’s brightness overwhelming the fluorescence emitted by the cells. Furthermore, the fracture of a plastic strut during cryosectioning resulted in small fragments that could be mistaken for cells. To overcome these limitations, future studies should explore alternative imaging techniques that can minimize the brightness caused by the plastic material [40]. Other types of plastic resins or variations of 3D printers should also be investigated [41]. However, it is important to note that printers utilizing laser printing may encounter similar issues with fluorescent resins, as the resin requires light sensitivity for solidification. It should also be noted that the plastic made it harder to cryosection the samples. The plastic would sometimes not be cleanly sliced which would give less support to the cryogel and cause the cryogel to break upon sectioning. Additionally, the plastic occasionally fractured which would also increase the chance that the section would not cut cleanly.

Overall, incorporation of a 3D-printed lattice within a cryogel scaffold did not appear to adversely affect pore formation in the infilled cryogel. The combined scaffolds were also able to swell to their full potential within four minutes and maintain their size, despite the addition of the plastic gyroid. The 10% and 20% combined scaffolds exhibited a slight decrease in average swelling ratio, while the 30% combined scaffold exhibited a significant decrease. Additionally, all combined scaffolds demonstrated significantly lower absolute swelling capacity when compared to control cryogel. The incorporation of the lattice significantly increased the compressive modulus when compared to control cryogel. Finally, the 20% solidity combined scaffolds were seeded with bone osteosarcoma-derived cells, which resulted in a high level of cell infiltration within the first 5 days followed by some cellular death observed in all scaffolds after day 7.

## 5. Conclusions

It is desirable to develop an alternative to bone grafts for the treatment of complex critical-size defects [6]. Cryogels are an ideal scaffold for this application due to their macroporous structure and biocompatible properties, although their mechanical strength is relatively weak [14]. In contrast, 3D-printed scaffolds offer excellent mechanical properties, but current technology often has limited porosity resolution [24]. The ideal bone tissue engineering scaffold should combine the advantages of these two scaffold types while minimizing their deficiencies. This study aimed to achieve this balance by combining a 3D-printed plastic gyroid lattice scaffold with a chitosan–gelatin cryogel scaffold. The incorporation of the 3D-printed lattice led to several benefits over the standard control cryogel, including proof of concept for patient-specific geometry and significant increases in mechanical strength. The scaffolds also exhibited a similar porosity and swelling ratio to control cryogels. However, these combined scaffolds exhibit decreased elasticity and absolute swelling capacity and may be cytotoxic to cells. The 20% solid combined scaffolds were identified as the most advantageous due to their optimal swelling ratio and mechanical properties. Additionally, the 2.5 mm pore lattice was found to possess optimal mechanical strength and swelling capacity. However, the range of different lattice variables suggests that different parameters could be chosen to apply to different areas of bone such as cortical vs. trabecular bone or in different types of loading such as transverse vs. longitudinal [42]. This study demonstrated a promising method for combining two different types of scaffolds; however, future studies should explore using different 3D printing materials for improved biocompatibility and bone cell differentiation [43]. Additionally, patient-specific scans should be used to test geometry capabilities, and degradation rates should be assessed for future combined scaffold materials [13].

## 6. Patents

Provisional patent application submitted.

## Figures and Tables

**Figure 1 bioengineering-10-00889-f001:**
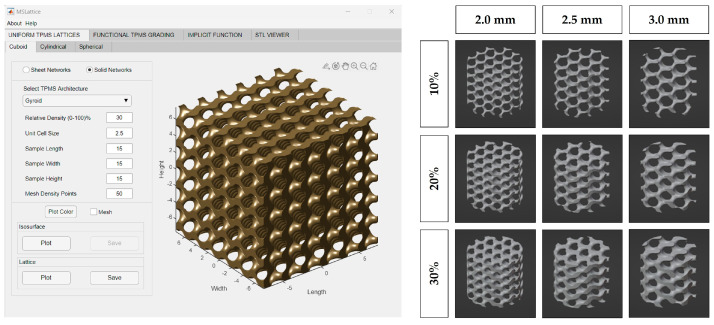
MSLattice interface with generated cuboidal gyroid (**left**). The 3D renderings of gyroid cylinder samples (later referred to as 3D−printed lattices; **right**). Columns represent different gyroid pore sizes and rows represent different gyroid solidities (fill percentage).

**Figure 2 bioengineering-10-00889-f002:**
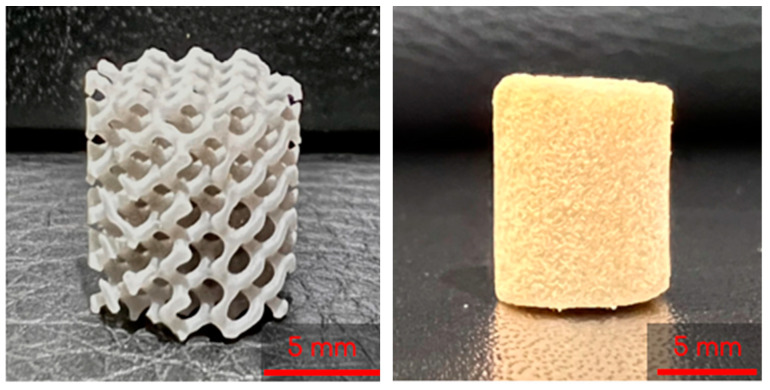
A 3D printed lattice (**left**) and lyophilized control cryogel (**right**).

**Figure 3 bioengineering-10-00889-f003:**
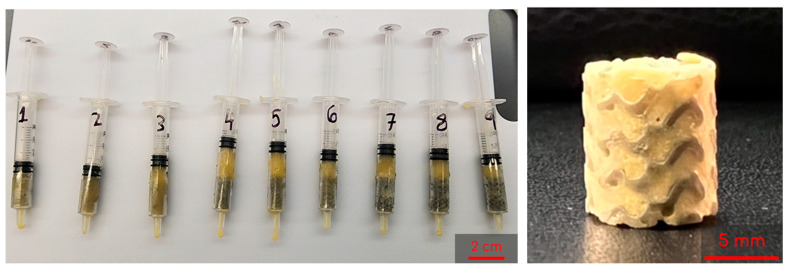
Example of combined scaffolds thawing in syringes after freezing for 18 h; syringes are numbered 1 through 9 as a shorthand for the 9 different lattices tested (**left**). Sample of a lyophilized combined scaffold (**right**).

**Figure 4 bioengineering-10-00889-f004:**
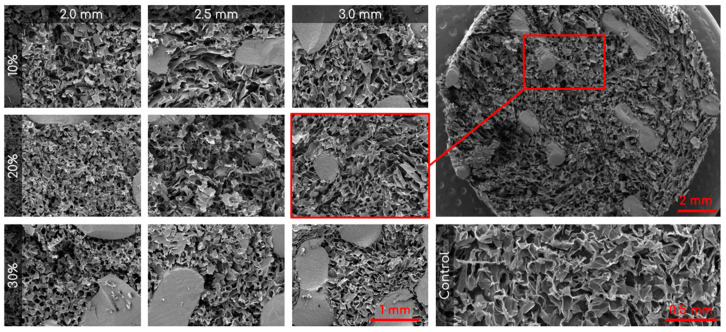
Representative SEM images of the nine different combined scaffolds (100×; columns represent lattice pore sizes and rows represent lattice solidity) and control cryogel (100×). Top right is a zoomed out visual of the combined scaffold (19×).

**Figure 5 bioengineering-10-00889-f005:**
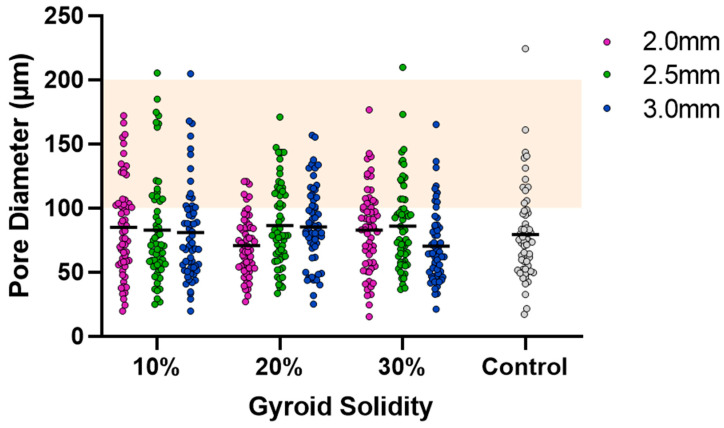
Cryogel pore analysis. Grey points are control values. Solid lines are mean values. Orange highlight is the range for ideal pore size (100–200 µm).

**Figure 6 bioengineering-10-00889-f006:**
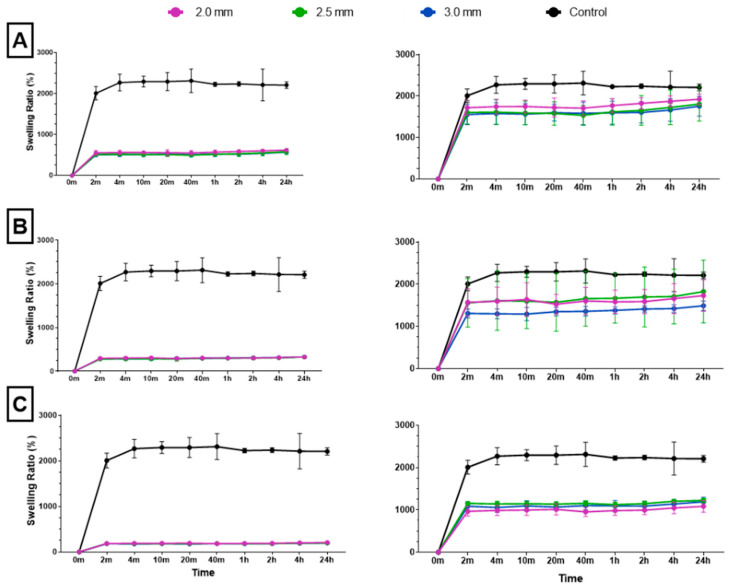
The (A) 10%, (B) 20%, and (C) 30% solidity combined scaffold absolute swelling ratios (left column) and relative swelling ratios (right column). Relative swelling ratios are with 3D-printed structure weights subtracted from total scaffold weight.

**Figure 7 bioengineering-10-00889-f007:**
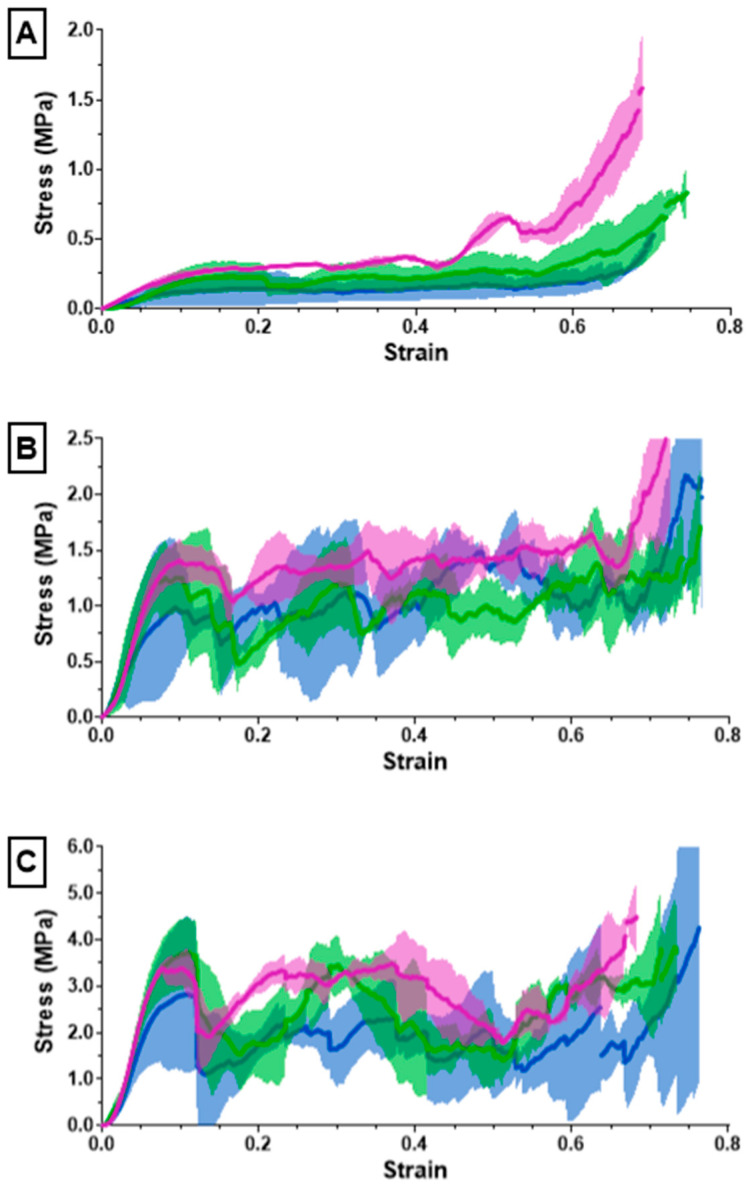
The (A) 10%, (B) 20%, (C) 30% solidity combined scaffold stress–strain results from compression testing using an Instron 68SC-2.

**Figure 8 bioengineering-10-00889-f008:**
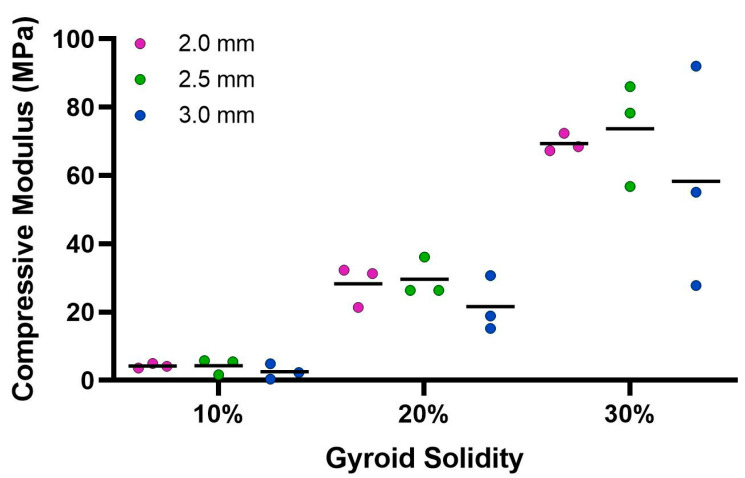
Average compressive modulus calculated from Instron 68SC-2 compression test. The control compressive modulus is 0.032 MPa.

**Figure 9 bioengineering-10-00889-f009:**
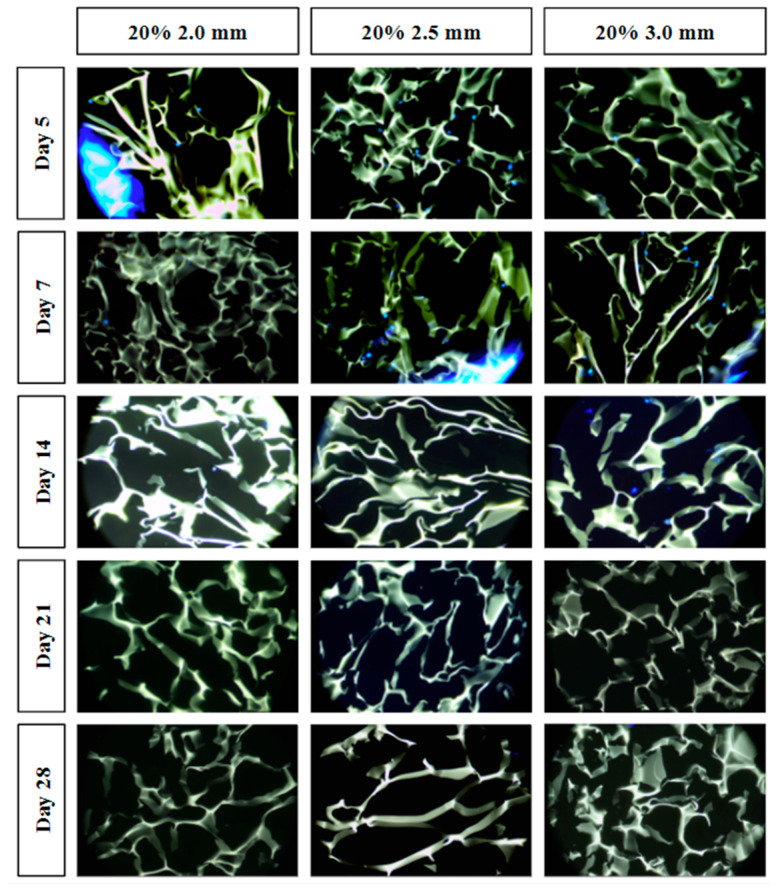
Cell infiltration stained images using DAPI staining and light microscopy. Individual fluorescent blue dots indicate the nuclei of the MG-63 cells. The green-yellow structures indicate the cryogel scaffold. Large blue fluorescence indicates the 3D-printed lattice (e.g., left side of the day 5 20% 2.0 mm combined scaffold).

**Table 1 bioengineering-10-00889-t001:** Averages and standard deviations for cryogel pore size inside the combined scaffolds (*n* = 60). Control mean is 79.27 μm and std. dev. is 35.86 μm.

	10% Solidity			20% Solidity			30% Solidity		
Gyroid Pore Size (mm)	2.0	2.5	3.0	2.0	2.5	3.0	2.0	2.5	3.0
Mean (μm)	84.90	82.85	80.90	70.82	86.27	85.38	82.73	86.01	70.22
Std. Dev. (μm)	37.74	41.70	37.41	23.13	32.66	31.09	32.83	34.84	28.35

**Table 2 bioengineering-10-00889-t002:** Compressive modulus for each combined scaffold.

	10% Solidity			20% Solidity			30% Solidity		
Gyroid Pore Size (mm)	2.0	2.5	3.0	2.0	2.5	3.0	2.0	2.5	3.0
Mean (MPa)	4.23	4.28	2.50	28.33	29.62	21.60	69.36	73.69	58.29
Std. Dev. (MPa)	0.66	2.34	2.26	6.04	5.62	8.08	2.67	15.16	32.24

## Data Availability

Not applicable.

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
