# Peer review of "Direct Integration of 3D Printing and Cryogel Scaffolds for Bone Tissue Engineering"

_bioengineering, 2023, doi:10.3390/bioengineering10080889_

Round 1

Reviewer 1 Report

This is very well written and interesting paper that investigates a practical problem when using cryogels in bone regeneration. The results are well discussed and the data are of high quality, i have the following minor comments for the authors. 

1. Figure 4 and Table 1 use the term gyroid solidity, this is not well introduced and the term relative density seems to be used in the materials and methods. Could the authors check for consistency in terminology and clarify their meaning for the reader - if appropriate.

2. in general the figure legends are not very descriptive, could the authors expand these to better describe the figures. For example, many figures have multiple parts that could be labelled a, b, c etc. in particular figure 4 has lots of different photos, are these from the same scaffold or different ones?

3. Figure 9 is not particularly clear, could the authors expand the legend - are the DAPI stained cells the green lines? without this information the figure is hard to understand.

4. lines 116/ 117 in the proof the authors should provide more details on the 3D print settings used, such as layer height, exposure time etc. this will enable other to reproduce the study.

5. it would be very helpful if the authors could provide .stl files in the final article as SI, so other users can replicate/ modify in future studies.

Reviewer 2 Report

The article is interesting and scientifically sound, some minor issue should be improved to make the paper more readable

The first part of the introduction should be revised as the english structure is not clear and (line 25-37)

The detail of 3d printing should be further expanded

The finding of the paper should be moved into discussion section and not into introduction where only the aim of the paper should be discussed

The 3d printer setting should be included in the material and method section

the fig 4 should indicate the magnification of each image

The A.A should compare and discuss the porosity and the macro and microstructure with the one of the natural bone Lo Giudice R, Puleio F, Rizzo D, Alibrandi A, Lo Giudice G, Centofanti A, et al. Comparative investigation of cutting devices on bone blocks: An SEM morphological analysis. Appl Sci 2019;9(2).

The english language is fine, but some paragraph should be improved to avoid repetition of terms and to be more clear

Reviewer 3 Report

Although this is a good experimental design, there are still many places that need to be modified, please follow the attached notes

Round 2

Reviewer 3 Report

The suggestion is acceptable as the author has successfully revised the manuscript.